# Accentuating the positive and eliminating the negative: Efficacy of TiO$_2$ as digestibility index marker for poultry nutrition studies

**Colleen Sprigg[1], Philip T. Leftwich[1], Emily Burton[2], Dawn Scholey[2], Michael R. Bedford[3], Charles A. Brearley[1]\***

**1** School of Biological Sciences, University of East Anglia, Norwich, United Kingdom, **2** School of Animal, Rural and Environmental Sciences, Nottingham Trent University, Southwell, United Kingdom, **3** AB Vista, Marlborough, Wiltshire, United Kingdom

\* c.brearley@uea.ac.uk

**Data Availability Statement:** All relevant data are within the manuscript and its Supporting Information files.

## Abstract

Inert digestibility index markers such as titanium dioxide are universally accepted to provide simple measurement of digestive tract retention and relative digestibility in poultry feeding trials. Their use underpins industry practice: specifically dosing regimens for adjunct enzymes added to animal feed. Among these, phytases, enzymes that degrade dietary phytate, inositol hexakisphosphate, represent a billion-dollar sector in an industry that raises ca. 70 billion chickens/annum. Unbeknown to the feed enzyme sector, is the growth in cell biology of use of titanium dioxide for enrichment of inositol phosphates from extracts of cells and tissues. The adoption of titanium dioxide in cell biology arises from its affinity under acid conditions for phosphates, suggesting that in feeding trial contexts that target phytate degradation this marker may not be as inert as assumed. We show that feed grade titanium dioxide enriches a mixed population of higher and lower inositol phosphates from acid solutions. Additionally, we compared the extractable inositol phosphates in gizzard and ileal digesta of 21day old male Ross 308 broilers fed three phytase doses (0, 500 and 6000 FTU/kg feed) and one inositol dose (2g/kg feed). This experiment was performed with or without titanium dioxide added as a digestibility index marker at a level of 0.5%, with all diets fed for 21 days. Analysis yielded no significant difference in effect of phytase inclusion in the presence or absence of titanium dioxide. Thus, despite the utility of titanium dioxide for recovery of inositol phosphates from biological samples, it seems that its use as an inert marker in digestibility trials is justified—as its inclusion in mash diets does not interfere with the recovery of inositol phosphates from digesta samples.

## Introduction

Inert digestibility index markers are the anchor of many feeding trials designed to analyse nutrient utilisation and amino acid digestibility along the gastrointestinal tract of animals. Marker use is widely accepted as a method less laborious than total tract retention.

**Funding:** This study was funded by award of a Biotechnology and Biological Sciences Research Council (BBSRC) Norwich Research Park Doctoral Training Studentship (Ref. BB/M011216/1) https://biodtp.norwichresearchpark.ac.uk/ to C.S. with contribution from AB Vista. BBSRC had no role in study design, data collection and analysis, decision to publish, or preparation of the manuscript.

**Competing interests:** The feeding trials described in this study were commissioned at Nottingham Trent University by AB Vista. AB Vista had no role in conduct of the trial, data collection and analysis, decision to publish or preparation of the manuscript. This does not alter our adherence to PLOS ONE policies on sharing data and materials.

Additionally, it enables the determination of digestibility along intestinal sections in ruminant and non-ruminant animals [1]. Digestibility markers, chromium dioxide, $CrO_2$, or chromic oxide, $Cr_2O_3$, titanium dioxide, $TiO_2$ and acid-insoluble ash are commonly used at inclusion level of 0.1 to 0.5%, though the choice of marker in individual studies is rarely qualified. A meta-analysis of poultry studies revealed substantive differences in ileal amino acid digestibility coefficients between $CrO_2$-, $TiO_2$- or acid-insoluble ash-normalized feeding trials [2]. A few studies have compared the use of chromic oxide ($Cr_2O_3$), a widely accepted digestibility index marker in ruminant animals, with the commonly used inert marker $TiO_2$ in monogastrics, for effect on apparent ileal digestibility of gross energy and nitrogen or recovery of the marker itself. The effects can be diet-dependent [3]. Nevertheless, the reproducibility of measurements of $TiO_2$, post recovery, and the safety of $TiO_2$ compared to the carcinogenic properties of $Cr_2O_3$, have positioned $TiO_2$ as the most suitable marker for use in most animal feeding trials. However, the interest in the comparison of different digestibility index markers [2,3] has brought to the fore the disparities in results with different markers which implies they may have matrix-specific and/or gut segment-specific interactions.

Quite separately the employ of $TiO_2$ has grown in a parallel field of research, intracellular cell biology, where it has been used as a solid phase extraction medium for concentration of inositol phosphates. The phosphate adsorbing properties of titanium were first documented in 1990 [4], but only in recent years Wilson and colleagues [5] reported how $TiO_2$ enabled enrichment of inositol phosphates, present at micromolar levels in cell lines and animal tissues, allowing their detection by polyacrylamide gel electrophoresis. Even before this, $TiO_2$ had been widely adopted for pre-purification of phosphopeptides for LC-MS analysis of protein phosphorylation [6]. More recently, extremely sensitive detection of inositol phosphates and inositol pyrophosphates has been achieved by combination of $TiO_2$-enrichment with mass spectrometry methods [7]. While there are remarkably few comparative studies of inositol phosphate speciation of different organs or effect of diet thereon, we recently described the use of $TiO_2$ as solid phase extraction medium with high performance liquid chromatography (HPLC)-based separations for measurements of inositol phosphates in poultry tissues [8]. The much higher levels of inositol phosphates in avian erythrocytes allow similar measurement without recourse to $TiO_2$ by employ of HPLC methods that are widely used in poultry nutrition studies for analysis of gizzard and ileal contents [9,10]. The use of $TiO_2$ to provide complementary inositol phosphate measurements in tissues and organs, beside more commonly reported measurements of inositol phosphate levels in digesta, provides opportunities to broaden understanding of animal response to dietary phytase supplementation. We may expect an extension of understanding of the benefits of phytases beyond the widely reported liberation of phosphate and inositol and the reduction of antinutritive properties of $InsP_6$ within the digestive tract [11]. The increasing industry practice of phytase 'super-dosing' (where phytase is added to feed at a dose that exceeds the standard level used to release phytate-bound P) has heightened the importance of using a marker capable of accurately measuring nutrient digestibility.

The use of $TiO_2$ to enrich inositol phosphates in cell biology [5] raised concerns over the potential interference of $TiO_2$ in inositol phosphate measurements in poultry digesta when $TiO_2$ is employed as a digestibility index marker. The employ of $TiO_2$ for purification of inositol phosphates and phosphopeptides requires incubation at an acidic pH for adsorption to the surface of the $TiO_2$ particles before subsequent elution at basic pH. These conditions closely resemble the transit time and pH changes witnessed by feed in different segments of the digestive tract of broiler chickens [12], raising the theoretical possibility that the digestive tract provides opportunity for interactions between $TiO_2$ as a marker and InsPs in the digestive tract. To confirm or deny the efficacy of the use of $TiO_2$ digestibility index marker in studies of

phytate digestion, this study aimed to investigate whether the inclusion of TiO$_2$ as a digestibility index marker at 0.5% in the diet of broilers had any interaction with supplemented phytase on the recovery of inositol phosphate species (and sum thereof) from digesta taken from gizzard and ileum. We also tested feed and 'cell biological' grade TiO$_2$ products for their ability to retain a mixed population of inositol phosphates, here obtained from an acid-hydrolysate of InsP$_6$. The two issues are, in combination, of relevance to discussion of the mechanisms by which phytase 'super-dosing' imparts benefits in poultry nutrition: much of the guidance offered for dosing of phytase is derived from measurements of the profiles of inositol phosphates recovered from specific segments of the gut [10,13–15].

## Materials and methods

### Care and use of animals

The study was performed at the Poultry Research Unit, School of Animal, Rural and Environmental Sciences, Nottingham Trent University (NTU). Ethical approval to conduct the study was obtained by the NTU animal ethics review committee (internal code ARE202134) and institutional and UK national NC3R ARRIVE guidelines for the care, use and reporting of animals in research were followed during the study. Birds had *ad libitum* access to feed and water throughout the study period.

### Animals, diet and experimental design

The investigation was carried out using 8 treatment groups, in which half of the diets were supplemented with 5 g/kg TiO$_2$ (Titanium, Ti) and the other half were not (Table 1). Within each Ti factor, diets were labelled as Control (no further supplementation), Ins (2 g/kg $^{12}$C/$^{13}$C inositol, containing $^{13}$C inositol at d30‰), Phy500 or Phy6000 where either 500 or 6000 FTU/kg phytase were added respectively. The phytase used was Quantum Blue, a thermo-tolerant modified *E. coli* 6-phytase (EC 3.1.3.26) supplied by AB Vista (Marlborough, UK). 480 male Ross 308 hatchlings were obtained from a commercial hatchery (PD Hook, Cote, Oxford, UK) and allocated randomly to 48 floor pens on day 1. For each of the 8 treatment groups, 10 birds were allocated to 6 replicate pens with birds in each pen fed one of the 8 diets for the duration of the trial (1 to 21 days). Data for mean gizzard and ileal inositol contents in responses to phytase addition with N and standard error reported by Walk *et al.* (2018) [16] was used to conduct a power calculation indicating 6 replicates per treatment were sufficient to identify treatment differences at a power setting of 80% and a type 1 error rate of 5%.

The composition of the basal diet is given in S1 Table. The basal diet was formulated to contain adequate levels of all nutrients according to the Ross Management Manual 2018. The 21-day study was made up of one diet phase–a starter–offered as a mash diet. The basal diets were supplemented or not with TiO$_2$ at 5g/kg. Animals, animal husbandry and basal diet composition of this trial have been described previously in an accompanying study of effect of dietary phytase on tissue inositol phosphates [8].

### Sampling and analytical methods

**Animals.**   Two birds per pen were randomly selected for sampling on d 21 post-hatch and were euthanised by cervical dislocation without prior stunning in accordance with the Welfare of Animals at the Time of Killing (England) Regulations (2015) guidelines for poultry. For each bird, the gizzard was excised, opened and the contents scraped into a container as a pooled sample from both birds. Ileal digesta were collected from the same two birds by gentle digital pressure, pooled and stored at -20˚C prior to freeze-drying, and were freeze dried at

**Table 1. Dietary treatments and Test Substance inclusion rates.**

| Dietary Treatment | Test Substance inclusion rates to the basal diet | | |
|---|---|---|---|
| | Quantum Blue g/tonne | [13]C Inositol mix g/tonne | Titanium Dioxide g/tonne |
| Control | - | - | - |
| 2 g/kg Ins | - | 2000 | - |
| Phy500 | 100 | - | - |
| Phy6000 | 1200 | - | - |
| Control Ti | - | - | 5000 |
| 2 g/kg Ins Ti | - | 2000 | 5000 |
| Phy500 Ti | 100 | - | 5000 |
| Phy6000 Ti | 1200 | - | 5000 |

-50°C for 7 days or until constant weight [17]. Once dried, samples were finely ground using a coffee grinder and stored at 4°C until analysis.

**Analysis of inositol phosphates in digesta.** Diets, gizzard and ileal digesta were extracted as described [8]. In brief, 100 mg samples of milled, dry feed or digesta were extracted in 5 mL of 100 mM NaF, 20 mM $Na_2$EDTA (pH 10) for 30 minutes shaking, followed by 30 minutes in a chilled bath sonicator and a further 2 hours standing at 4°C. The extract was centrifuged at 9000 x $g$ for 15 minutes at 4°C and 1 mL was filtered through a 13-mm 0.45μm PTFE syringe filter (Kinesis, UK). Aliquots (20 μL) were analysed by HPLC with UV detection at 290 nm after post-column complexation of inositol phosphates with ferric ion [8].

**Analysis of inositol phosphates recovered from TiO₂.** Comparison of recovery of mixed inositol phosphates from an acid-hydrolysate of $InsP_6$ by feed grade and 'cell biological' grade $TiO_2$ was carried out as described [8]. Briefly, 500 μL of 2 mM $InsP_6$, as hydrolysate (pH 1, in 1M $HClO_4$) was mixed with 5mg of either titanium dioxide ($TiO_2$) beads (Titansphere® $TiO_2$ 5 μM, Hichrom) or feed grade $TiO_2$ (Target Feeds Ltd, UK, particle size ranging 1 to 150 nm) and incubated at 4°C for 30 minutes with mixing on a rotator. Samples were centrifuged at 3500 x $g$ for 5 minutes to pellet the $TiO_2$ and the $HClO_4$ supernatant was discarded. Inositol phosphates bound to the $TiO_2$ beads were eluted by two 5-minute washes at 4°C in 200 μL 3% ammonium hydroxide solution (pH 10). The eluates were pooled, vacuum evaporated until dry and resuspended in 100 μL of 18.2 MOhm.cm water. Aliquots (20 μL) were analysed by HPLC with UV detection at 290 nm after post-column complexation of inositol phosphates with ferric ion [8]. The elution order of inositol phosphates was established using a set of standards prepared by acid-hydrolysis of $InsP_6$, with concentration established by reference to UV detector response to injection of $InsP_6$ (Merck).

## Statistical analysis

For all 8 treatments (Control, 2 g/kg inositol, 500 FTU/kg and 6000FTU/kg Quantum Blue phytase; with and without 5 g/kg $TiO_2$) inositol phosphates and their aggregated totals were compared by two-way ANOVA. The interaction term between diet/treatment and $TiO_2$ was removed in favour of an additive model as the sum of squared errors between the two models are not significantly different. All analyses carried out with R v 4.1.3 [18]. The level of significance for all tests was set at α ≤ 0.05.

## Results

To test whether feed grade $TiO_2$ (Target Feeds, UK) retains the same property for binding phosphate-containing compounds as Titansphere® $TiO_2$ used for phosphopeptide and

inositol phosphate purification and enrichment, the materials were compared for ability to capture a mixed population of inositol phosphates in an acid-hydrolysate of inositol hexaki-sphosphate (InsP$_6$) sample (Fig 1).

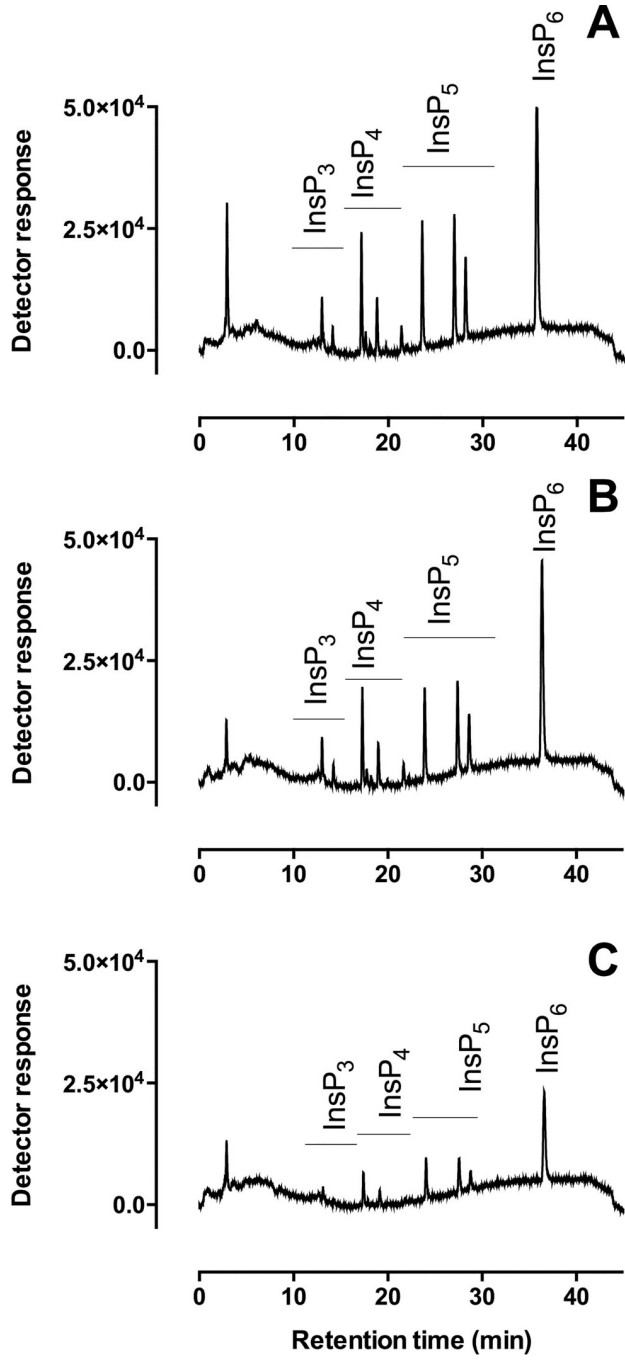

**Fig 1. Recovery of inositol phosphates using titanium dioxide.** (**A**) An acid hydrolysate of InsP$_6$ with HPLC-resolvable lower inositol phosphates, (**B**) Inositol phosphates recovered from 'cell biological' grade Titansphere TiO$_2$, (**C**) Inositol phosphates recovered from feed grade TiO$_2$. The retention times of the different classes of inositol phosphates are shown.

With comparison to the known spiked concentration of an inositol hexakisphosphate hydrolysate used as a model sample, 'cell biological' grade Titansphere® TiO$_2$ showed greater than twice the binding capacity of the feed grade TiO$_2$, with calculated recovery of inositol phosphates following extraction of 79% and 32%, respectively. The ability of the feed grade TiO$_2$ to bind inositol phosphates, recovering almost identical profiles of inositol phosphates therefrom, suggests that the material used is not as inert as considered. It retains the chemical ability to interact with phosphate-containing compounds under acid conditions that might be experienced *in vivo* in the gizzard.

The previously published extraction of inositol phosphates using TiO$_2$ [5,8] employs incubation at an acidic pH for an extended period for TiO$_2$ to bind phosphate-containing compounds before subsequent brief elution of these at basic pH. This transition from binding in acidic to elution in basic conditions mirrors the transit time [19] and pH changes measured in different segments of the digestive tract of broiler chickens, from acidic pH (2.5–3.5) in the proventriculus/gizzard to neutral to mildly basic in the terminal ileum and cecum/colon (pH 6–8) [12], though more extreme conditions have been monitored *in vivo* in gizzard [19]. Clearly, these bolus-witnessed pH changes raise the possibility that dietary phytate and released inositol phosphates bind to TiO$_2$ in the gizzard and may be partially eluted therefrom in distal parts of the gut where pH reaches more basic conditions. By corollary, we might expect complex interactions between feed phytate, calcium, phytase dose, TiO$_2$ and gut segment. To test the potential interaction of feed grade TiO$_2$ with inositol phosphates in the gut lumen, we compared extractable inositol phosphate concentrations from gizzard (a low pH segment) and ileal (a higher pH segment) digesta of birds fed identical diets with and without the presence of 5g/kg TiO$_2$ as a marker and phytase at three doses (Tables 2 and 3). We also included inositol at a level anticipated of complete dephosphorylation of phytate.

Inositol phosphates present in gizzard digesta of d21 broilers (Table 2) were extracted and measured in replicate treatment groups with and without 5 g/kg TiO$_2$ as a digestibility index marker, to ascertain whether its inclusion impacted the analysis of inositol phosphates. Analysis by two-way ANOVA showed no significant interaction of the factors, diet and titanium ($F_{3,40} = 0.744$, $p = 0.5322$), with interaction accounting for only 0.99% of the total variance. The analysis was repeated, removing interaction from the model, but we found no evidence

**Table 2. Inositol phosphate (InsP$_{2-6}$) levels (nmol/g dwt) in gizzard digesta of day 21 broilers[1,2].**

| Diet | InsP$_2$ | InsP$_3$ | InsP$_4$ | InsP$_5$ | InsP$_6$ | Σ InsP |
|---|---|---|---|---|---|---|
| Control | 193±42 | 591±137 | 2474±485 | 4722±441 | 6873±996 | 14852±817 |
| Control Ti | 220±27 | 806±221 | 3586±1196 | 3249±545 | 4548±1540 | 12410±720 |
| 2g/kg Ins | 167±27 | 527±159 | 2368±752 | 4462±469 | 6483±1218 | 14008±884 |
| 2g/kg Ins Ti | 154±36 | 670±306 | 2836±1160 | 3625±460 | 5959±2076 | 13245±1220 |
| Phy500 | 677±139 | 1777±728 | 3387±1354 | 1122±419 | 1645±907 | 8606±1757 |
| Phy500 Ti | 546±126 | 2593±535 | 3788±850 | 906±572 | 1426±1065 | 9257±1676 |
| Phy6000 | 75±22 | 417±166 | 299±81 | 97±61 | 142±57 | 1030±183 |
| Phy6000 Ti | 117±39 | 344±54 | 149±73 | 101±46 | 239±39 | 950±68 |

Abbreviations: Σ InsP, total InsP$_2$ to InsP$_6$; InsP$_6$, inositol hexakisphosphate; InsP$_5$, inositol pentakisphosphate; InsP$_4$, inositol tetrakisphosphate; InsP$_3$, inositol trisphosphate; InsP$_2$, inositol bisphosphate; Ti, TiO$_2$.

[1]The control group was fed with a diet with 0.45% calculated available phosphate. Groups 2g/kg Ins were fed with 2g supplemented d30‰ [13]C inositol mix per kilogram of feed; groups Phy500 and Phy6000 were fed with the control diet supplemented with 500 or 6,000 FTU of phytase per kilogram of feed, respectively. Ti groups had 5g/kg TiO$_2$ added to the diet as a digestibility index marker.

[2]Data are given as group means ± SEM, n = 6, of 6 replicate pens with samples pooled from 2 broilers per pen per treatment.

for an effect of TiO$_2$ on total extractable inositol phosphates in the gizzard digesta ($F_{1,43}$ = 0.75, $p$ = 0.39), accounting for only 0.40% of total variance, despite numerical differences between paired treatment groups with and without titanium. Dietary treatment by provision of phytase was found to be a significant source of explained variance ($F_{3,43}$ = 61.7, $p$ < 0.001), and the significant reduction in total inositol phosphates in the gizzard digesta at inclusion of these phytase levels has been previously reported [8].

As presented above for gizzard digesta (Table 2), inositol phosphates were measured in ileal digesta (Table 3) at graded phytase levels with and without titanium dioxide as a digestibility index marker. The data was compared by two-way ANOVA to determine whether inclusion of TiO$_2$ was a source of variance in the measurable inositol phosphates. Interaction between diet (phytase inclusion, inositol or control) and inclusion of TiO$_2$ accounted for only 1.89% of the total variance, and the interaction was considered not significant ($F_{3,40}$ = 2.209, $p$ = 0.1019). Analysis of total extractable inositol phosphates in the ileal digesta by two-way ANOVA showed no significant effect of TiO$_2$ on the recovery of mixed inositol phosphates ($F_{1,43}$ = 0.71, $p$ = 0.4), whereas dietary treatment by way of inclusion of phytase was highly significant ($F_{3,43}$ = 93.51, $p$ < 0.001). The effect of phytase in significantly reducing inositol phosphates measured in the ileal digesta of d21 broilers in the treatment groups presented here has been described [8].

## Discussion

While TiO$_2$ has been used in cell biology for phosphopeptide research for many years [4,6], its recent employ as a pre-concentration matrix for the extraction of inositol phosphates [5,8] was tested in an animal nutrition context, poultry, only recently [8]. In contrast, use of TiO$_2$ as a digestibility index marker in the animal feed industry is longstanding [3]. A recent study conducted a meta-analysis of open science practice in animal research [20] and cited the reluctance and/or difficulty in publishing 'negative' or null-hypothesis results as a structural problem of scientific publishing, particularly in animal research. While TiO$_2$ remains a digestibility index marker of choice in animal feeding trials in poultry and swine following studies showing its ease of measurement and reduced risk compared to Cr [21,22], supported by a

**Table 3. Inositol phosphate (InsP$_{2-6}$) levels (nmol/g dwt) in ileal digesta of day 21 broilers[1,2].**

| Diet | InsP$_2$ | InsP$_3$ | InsP$_4$ | InsP$_5$ | InsP$_6$ | Σ InsP |
|---|---|---|---|---|---|---|
| Control | 502±131 | 1358±156 | 2613±307 | 5285±519 | 51588±3269 | 61347±3702 |
| Control Ti | 523±93 | 1270±114 | 2315±259 | 4683±447 | 48564±3126 | 57356±3599 |
| 2g/kg Ins | 656±122 | 1610±224 | 2711±265 | 5715±409 | 56214±2966 | 66905±3306 |
| 2g/kg Ins Ti | 487±138 | 1371±181 | 2388±224 | 4738±379 | 47216±2585 | 56145±2940 |
| Phy500 | 1349±173 | 3401±584 | 7038±2014 | 6199±682 | 29190±3804 | 40411±3922 |
| Phy500 Ti | 3338±2200 | 3250±515 | 6596±1734 | 6045±403 | 24865±2489 | 44094±2938 |
| Phy6000 | 2536±660 | 1614±358 | 3843±1173 | 433±97 | 1748±386 | 10174±2236 |
| Phy6000 Ti | 2257±565 | 2169±484 | 4760±1361 | 819±308 | 3217±1388 | 13221±2886 |

Abbreviations: Σ InsP, total InsP$_2$ to InsP$_6$; InsP$_6$, inositol hexakisphosphate; InsP$_5$, inositol pentakisphosphate; InsP$_4$, inositol tetrakisphosphate; InsP$_3$, inositol trisphosphate; InsP$_2$, inositol bisphosphate; Ti, TiO$_2$.

[1]The control group was fed with a diet with 0.45% calculated available phosphate. Groups 2g/kg Ins were fed with 2g supplemented d30‰ $^{13}$C inositol mix per kilogram of feed; groups Phy500 and Phy6000 were fed with the control diet supplemented with 500 or 6,000 FTU of phytase per kilogram of feed, respectively. Ti groups had 5g/kg TiO$_2$ added to the diet as a digestibility index marker.

[2]Data are given as group means ± SEM, n = 6, of 6 replicate pens with samples pooled from 2 broilers per pen per treatment.

body of global feeding trial studies containing its use [reviewed, 3], studies such as this may offer reassurance of the efficacy of approach in reporting this 'negative' result. The boundaries of improved animal performance gained by phytase addition remain those revealed in critical meta-analysis [2]. We conclude that despite the demonstrable ability of feed grade $TiO_2$ to bind inositol phosphates under acidic conditions, in this study the activity of adjunct phytase, as measurable soluble total inositol phosphates in digesta, was not impacted by $TiO_2$ at an inclusion rate of 5 g/kg $TiO_2$, while the reduction of InsPs in digesta with phytase inclusion followed the pattern reported in previous studies [10,11,13–15]. The same may or may not hold true for different feed matrices, for other digestibility index markers or for swine [22,23]. Even so, phytases continue to provide resource-efficiencies for sustainable animal production. The data presented here vouchsafes historic use of $TiO_2$ as digestibility index marker and its use in contemporary feeding trials where tissue inositol phosphate analysis is itself enabled by use of $TiO_2$ to enrich inositol phosphates before analysis [8]. In short, $TiO_2$ may be used to further elucidate tissue-specific mechanisms by which phytase benefits animal performance.

## Supporting information

**S1 Table. Ingredient composition and calculated nutrient concentrations of the basal diet.** (DOCX)

**S1 Data. Spreadsheet of underlying data for tables.** (XLSX)

## Acknowledgments

We acknowledge the technical assistance of staff of the Nottingham Trent University Poultry Research Unit.

## Author Contributions

**Conceptualization:** Emily Burton, Michael R. Bedford, Charles A. Brearley.

**Data curation:** Philip T. Leftwich.

**Formal analysis:** Philip T. Leftwich.

**Funding acquisition:** Michael R. Bedford, Charles A. Brearley.

**Investigation:** Colleen Sprigg, Emily Burton, Dawn Scholey.

**Methodology:** Colleen Sprigg, Emily Burton, Dawn Scholey.

**Project administration:** Emily Burton, Dawn Scholey, Michael R. Bedford.

**Software:** Philip T. Leftwich.

**Supervision:** Charles A. Brearley.

**Writing – original draft:** Colleen Sprigg, Charles A. Brearley.

**Writing – review & editing:** Philip T. Leftwich, Charles A. Brearley.

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
