## [Decision Letter · Decision Letter 0]

4 Dec 2022

PONE-D-22-30456Accentuating the positive and eliminating the negative: efficacy of TiO_2_ as digestibility index marker for poultry nutrition studies__PLOS ONE

Dear Dr. Brearley,

Thank you for submitting your manuscript to PLOS ONE. After careful consideration, we feel that it has merit but does not fully meet PLOS ONE’s publication criteria as it currently stands. Therefore, we invite you to submit a revised version of the manuscript that addresses the points raised during the review process. Dear Authros

I believe the experiment is carefully design, however, I and one of the reviewer is confused about your experimental design and analysis. For instance, readers don’t need to go to the accompanying paper to understand the experiment’s setup, therefore, should be explained in text. I don’t see a 2x4 in this description. It doesn’t mean it wasn’t a 2x4 arrangement, but it needs to be fully explained. What are the factors? If, for instance, we learn that the treatment arrangement was a 2x4. It begs the question of why the data in the tables were not analyzed as a 2x4. We hope you will consider our request in the revised manuscript. 

We look forward to receiving your revised manuscript.

Kind regards,

Aziz ur Rahman Muhammad

Academic Editor

PLOS ONE

“The feeding trials described in this study were commissioned at Nottingham Trent University by AB Vista.  AB Vista had no role in conduct of the trial, data collection and analysis, decision to publish or preparation of the manuscript.”

Additional Editor Comments:

I believe the experiment is carefully design, however, I and one of the reviewer is confused about your experimental design and analysis. For instance, readers don’t need to go to the accompanying paper to understand the experiment’s setup, therefore, should be explained in text. I don’t see a 2x4 in this description. It doesn’t mean it wasn’t a 2x4 arrangement, but it needs to be fully explained. What are the factors? If, for instance, we learn that the treatment arrangement was a 2x4. It begs the question of why the data in the tables were not analyzed as a 2x4. We hope you will consider our request in the revised manuscript.

Reviewers' comments:

Reviewer's Responses to Questions

**Comments to the Author**

1. Is the manuscript technically sound, and do the data support the conclusions?

Reviewer #1: Yes

Reviewer #2: Yes

Reviewer #3: Yes

2. Has the statistical analysis been performed appropriately and rigorously? 

Reviewer #1: No

Reviewer #2: Yes

Reviewer #3: Yes

3. Have the authors made all data underlying the findings in their manuscript fully available?

Reviewer #1: Yes

Reviewer #2: Yes

Reviewer #3: Yes

4. Is the manuscript presented in an intelligible fashion and written in standard English?

Reviewer #1: Yes

Reviewer #2: Yes

Reviewer #3: Yes

5. Review Comments to the Author

Reviewer #1: The work is interesting but there are too many issues with the premise and presentation of the work. Specific comments are shown below.

L16: Facile? I suggest the authors use another word. That makes the whole exercise superficial and meaningless.

L16: total tract digestibility is a wrong word choice for several reasons. 1: when it is the total tract for poultry, it is not digestibility but retention. 2: index method is used not only for total tract…I suggest the authors simply state that it is used for digestibility and retention…

L19: what is $b

L19-26: this section is anticlimactic. Is it really necessary? The authors may consider rewording it, so it is less disappointing in the end.

L40-44: this premise may need to be adjusted. The index method is less laborious than total tract retention. Still, the use of markers also enables the determination of digestibility in the intestine, along intestinal sections, etc. lastly, the issue with poultry is not the contribution of microbial AA in the hindgut. This is applicable in other non-ruminants as well.

L49-52: I think this is more suitable for the discussion section, and also must be substantiated with appropriate refs

L87-90: this reads like a conclusion of the study. This should be deleted and moved to its appropriate place in the manuscript.

L93: Target Feeds is not the manufacturer of TiO2; the authors should cite the manufacturer

Fig 1: why are the InsP shown in panel B but not the others?

Tables 1 and 2: tables are too busy and very cumbersome to read. I suggest 1. That the authors replace the within samples standard errors with pooled standard errors. 2. Instead of providing these high p values that were not significantly different, write “NS,” defined below the table as not significant.

L178-81: I struggled hard to see how this fits in with the subject that this manuscript is trying to address. I suggest that the authors delete this because it detracts from their work.

L187-8: the point being made by the author is clear to me, but I don’t think that what the author claimed they found is what they found (or even investigated). The authors studied the recovery of InsP in diets with different Ti sources and found that TiO2 did not influence the InsP recovery. That is good to hear and shows that the old wisdom is not outdated (for now). However, the authors did not study (at least I didn’t see it) the effect of phytase, as in whether TiO2 affected the impact of phytase on InsP recovery. I see treatments with phytase (with or without Ti) – in this case, it seems that the subject is Ti, not phytase (in as much as both treatments have phytase). To show the interaction of phytase and Ti, there should at least be one more treatment (with Ti but without phytase). Or perhaps, the authors need to do a better job of explaining what they observed.

L190-5: I don’t see much in this conclusion. This reads more like a policy paper than the conclusion of rigorous scientific work. I like the work done here, but I don’t think the writing has done justice to the job done.

L213-5: I appeal to the authors to help me here. I don’t need to go to the accompanying paper to understand the experiment’s setup. I don’t see a 2x4 in this description. It doesn’t mean it wasn’t a 2x4 arrangement, but it needs to be fully explained. What are the factors?

L213-5: now that we learn that the treatment arrangement was a 2x4. It begs the question of why the data in the tables were not analyzed as a 2x4.

L232: finely ground is not defined. Be more specific, please.

L261-7: I didn’t see any evidence of Tukey separation in the tables presented here. But a more significant problem is using the T test to separate the means. Isn’t that what Tukey is supposed to do? Did the authors consider that the T-test is not used when multiple treatments are involved because there is a loss in statistical power with each sequence of comparisons? The correct thing to do would be to have analyzed the data as a 2x4 (in as much as that was the treatment arrangement). Unless, of course, I missed something.

Reviewer #2: This is a nice study which it worth to be known by the animal nutritionist that usually trust on digestibility values estimated by the index marker ratio, using TiO2 as one of the most used markers. The hypothesis is clear and well justified, and the experimental protocol allows to identify, if they would have existed, likely interference or interactions on the phytase activity in digesta.

There are only one minor comment that probably it has a proper explanation

Line 43.- Please, explain or delete the following statement “particularly in poultry where the contribution of microbial amino acids also confounds direct balance studies” Which is it the microbial interference that affects to direct studies and not to the marker index ratio estimate.

Reviewer #3: The manuscript “Accentuating the positive and eliminating the negative: "Efficacy of TiO2 as a digestibility index marker for poultry nutrition studies" presented by Sprigg et al, required a review. Manuscript included important information to support the proposal that the use of TiO2 as inert in the phosphorus digestibility trials, especially with enzyme evaluations. Are necessary corrections and clarifications to paper be accepted.

6. PLOS authors have the option to publish the peer review history of their article (what does this mean?). If published, this will include your full peer review and any attached files.

Reviewer #1: **Yes: **Oluyinka Abiona Olukosi

Reviewer #2: No

Reviewer #3: No

---

## [Author Response · Author response to Decision Letter 0]

21 Dec 2022

21 December 2022

Dear Aziz ur Rahman Muhammad

Re. Accentuating the positive and eliminating the negative: efficacy of TiO2 as digestibility index marker for poultry nutrition studies

We thank you for the opportunity to return a revised manuscript which we consider to be much improved by address to the criticisms raised in the review process. We thank our reviewers.

and

Confirmed

Our Competing Interests section now reads, “The feeding trials described in this study were commissioned at Nottingham Trent University by AB Vista. AB Vista had no role in conduct of the trial, data collection and analysis, decision to publish or preparation of the manuscript. This does not alter our adherence to PLOS ONE policies on sharing data and materials”

Additional Editor Comments:

I believe the experiment is carefully design, however, I and one of the reviewer is confused about your experimental design and analysis. For instance, readers don’t need to go to the accompanying paper to understand the experiment’s setup, therefore, should be explained in text. I don’t see a 2x4 in this description. It doesn’t mean it wasn’t a 2x4 arrangement, but it needs to be fully explained. What are the factors? If, for instance, we learn that the treatment arrangement was a 2x4. It begs the question of why the data in the tables were not analyzed as a 2x4. We hope you will consider our request in the revised manuscript.

We have altered the description of the trial design L26-30, L218-234 to make clear that the trial was designed to study the effect of titanium dioxide (TiO2) inclusion in diets with/without phytase. The tables of results and associated description L145-178 now include ANOVA to show effect of TiO2 (Titanium), diet and their interaction.

5. Review Comments to the Author

Reviewer #1: The work is interesting but there are too many issues with the premise and presentation of the work. Specific comments are shown below.

L16: Facile? I suggest the authors use another word. That makes the whole exercise superficial and meaningless.

Facile is replaced with ‘simple’

L16: total tract digestibility is a wrong word choice for several reasons. 1: when it is the total tract for poultry, it is not digestibility but retention. 2: index method is used not only for total tract…I suggest the authors simply state that it is used for digestibility and retention…

We stand corrected and have followed the suggestion with the sentence reading, ‘..of digestive tract retention and relative digestibility ..’.

L19: what is $b

Replaced with, ‘.. a billion dollar sector..’.

L19-26: this section is anticlimactic. Is it really necessary? The authors may consider rewording it, so it is less disappointing in the end.

We have retained the overall form of the Abstract to state our two main conclusions: those of new L24-26 and L31-34. We have, however, removed the ‘flowery’ sentence of the original L24-25, ‘ Consequently, the guiding tenets ..”.

L40-44: this premise may need to be adjusted. The index method is less laborious than total tract retention. Still, the use of markers also enables the determination of digestibility in the intestine, along intestinal sections, etc. lastly, the issue with poultry is not the contribution of microbial AA in the hindgut. This is applicable in other non-ruminants as well.

The reviewer is correct, we have modified the sentences accordingly.

L49-52: I think this is more suitable for the discussion section, and also must be substantiated with appropriate refs

We have left this aspect of the Introduction as written, because it sets the scene, but we have added extra references [new 23, 24, to pre-existing 2,3,4 and 22] and have provided a slightly more expansive discussion to the Discussion L183-199.

L87-90: this reads like a conclusion of the study. This should be deleted and moved to its appropriate place in the manuscript.

The text of L87-90 has been replaced with sentences new L98-104 that do not pre-empt the conclusions of the study.

L93: Target Feeds is not the manufacturer of TiO2; the authors should cite the manufacturer

We have scoured the poultry literature for statements of source of TiO2 used in feeding trials and can barely find any references to the source. Our source was Target Feeds, who have discontinued this product and no longer list it. We can offer no more information. 

Fig 1: why are the InsP shown in panel B but not the others?

The labels have been added to all panels

Tables 1 and 2: tables are too busy and very cumbersome to read. I suggest 1. That the authors replace the within samples standard errors with pooled standard errors. 2. Instead of providing these high p values that were not significantly different, write “NS,” defined below the table as not significant.

The tables have been simplified. They have been modified to provide ANOVA for the eight diets. We have done so to address the principal criticisms of the manuscript – the explicit description of trial design. The description of the statistical analysis has been modified accordingly, new L272-276.

L178-81: I struggled hard to see how this fits in with the subject that this manuscript is trying to address. I suggest that the authors delete this because it detracts from their work.

We can see where the reviewer with a poultry science perspective is coming from. While much poultry research is very close to application, an awful lot of cell biological work on inositol phosphates is rather more distant from its translation. This paper brings the two together, we expect this to be read by animal nutritionists and cell biologists alike. For this reason and specifically because of the open science mission of PLOS ONE, we prefer to retain it.

L187-8: the point being made by the author is clear to me, but I don’t think that what the author claimed they found is what they found (or even investigated). The authors studied the recovery of InsP in diets with different Ti sources and found that TiO2 did not influence the InsP recovery. That is good to hear and shows that the old wisdom is not outdated (for now). However, the authors did not study (at least I didn’t see it) the effect of phytase, as in whether TiO2 affected the impact of phytase on InsP recovery. I see treatments with phytase (with or without Ti) – in this case, it seems that the subject is Ti, not phytase (in as much as both treatments have phytase). To show the interaction of phytase and Ti, there should at least be one more treatment (with Ti but without phytase). Or perhaps, the authors need to do a better job of explaining what they observed.

We take the reviewer’s point that our description of the trial design could have been better and that we did not draw out of our analysis a description of the effect of phytase, in part because it has been described many times before. We have done so now: the modified Tables 1 and 2 retain all the original measured values (mean and SEM) but now include the ANOVA to reveal lack of interaction of titanium dioxide (Titanium) and diet (phytase), lack of effect of Titanium and effect of diet (phytase). The description of the statistical analysis has been modified accordingly, new L272-276.

L190-5: I don’t see much in this conclusion. This reads more like a policy paper than the conclusion of rigorous scientific work. I like the work done here, but I don’t think the writing has done justice to the job done.

We have modified the closing sentences of the discussion, new L193-197 to put our work in context of other phytase studies (ref 10, 11, 13-15). We have reduced the ‘policy’ aspect of the discussion and finish up, new L197-204, with a statement encouraging others to use TiO2 both as marker and as an aid to extraction of inositol phosphates from tissues of poultry. We cite [8], the first study to use TiO2 for the latter purpose.

L213-5: I appeal to the authors to help me here. I don’t need to go to the accompanying paper to understand the experiment’s setup. I don’t see a 2x4 in this description. It doesn’t mean it wasn’t a 2x4 arrangement, but it needs to be fully explained. What are the factors?

We hope we have provided greater clarity in our description of the design of the trial and in the presentation of the results with associated ANOVA Tables 1 and 2, which specifically identifies the factors, Titanium and diet. The description of the statistical analysis has been modified accordingly, new L272-276.

L213-5: now that we learn that the treatment arrangement was a 2x4. It begs the question of why the data in the tables were not analyzed as a 2x4.

We hope we have provided greater clarity in our description of the design of the trial and in the presentation of the results with associated ANOVA Tables 1and 2, which specifically identifies the factors, Titanium and diet. The description of the statistical analysis has been modified accordingly, new L272-276.

L232: finely ground is not defined. Be more specific, please.

We have added detail of the use of a coffee grinder, albeit without particle size analysis, and have provided further detail of the freeze-drying with reference [25] to the AOAC method, new L242-245.

L261-7: I didn’t see any evidence of Tukey separation in the tables presented here. But a more significant problem is using the T test to separate the means. Isn’t that what Tukey is supposed to do? Did the authors consider that the T-test is not used when multiple treatments are involved because there is a loss in statistical power with each sequence of comparisons? The correct thing to do would be to have analyzed the data as a 2x4 (in as much as that was the treatment arrangement). Unless, of course, I missed something.

The reviewer is correct, we have provided the ANOVA for clarity. The description of the statistical analysis has been modified accordingly, new L272-276.

Reviewer #2: This is a nice study which it worth to be known by the animal nutritionist that usually trust on digestibility values estimated by the index marker ratio, using TiO2 as one of the most used markers. The hypothesis is clear and well justified, and the experimental protocol allows to identify, if they would have existed, likely interference or interactions on the phytase activity in digesta.

We thank the reviewer.

There are only one minor comment that probably it has a proper explanation

Line 43.- Please, explain or delete the following statement “particularly in poultry where the contribution of microbial amino acids also confounds direct balance studies” Which is it the microbial interference that affects to direct studies and not to the marker index ratio estimate.

We have removed the statement.

Finally, we thank our reviewers and editor for their consideration of our manuscript.

For the purpose of open access, the author has applied a Creative Commons Attribution (CC BY) to any Author Accepted Manuscript version arising.

Yours sincerely

Charles Brearley

---

## [Decision Letter · Decision Letter 1]

7 Feb 2023

PONE-D-22-30456R1Accentuating the positive and eliminating the negative: efficacy of TiO2 as digestibility index marker for poultry nutrition studiesPLOS ONE

Dear Dr. Brearley,

Thank you for submitting your manuscript to PLOS ONE. After careful consideration, we feel that it has merit but does not fully meet PLOS ONE’s publication criteria as it currently stands. Therefore, we invite you to submit a revised version of the manuscript that addresses the points raised during the review process.

Dear Authors, 

I still believe that manuscript require to improve the statistical part as suggested by reviewer. after improving statistical part, manuscript will be ready for publication. I invite authors to address the issues of reviewer please. 

We look forward to receiving your revised manuscript.

Kind regards,

Aziz ur Rahman Muhammad

Academic Editor

PLOS ONE

Additional Editor Comments (if provided):

Dear Authors,

I still believe that authors are required to improve the statistical part as suggested by reviewer. after improving statistical part, manuscript will be ready for publication. I invite authors to address the issues of reviewer please.

Reviewers' comments:

Reviewer's Responses to Questions

**Comments to the Author**

1. If the authors have adequately addressed your comments raised in a previous round of review and you feel that this manuscript is now acceptable for publication, you may indicate that here to bypass the “Comments to the Author” section, enter your conflict of interest statement in the “Confidential to Editor” section, and submit your "Accept" recommendation.

Reviewer #1: (No Response)

2. Is the manuscript technically sound, and do the data support the conclusions?

Reviewer #1: Partly

3. Has the statistical analysis been performed appropriately and rigorously? 

Reviewer #1: No

4. Have the authors made all data underlying the findings in their manuscript fully available?

Reviewer #1: Yes

5. Is the manuscript presented in an intelligible fashion and written in standard English?

Reviewer #1: No

6. Review Comments to the Author

Reviewer #1: 1. I still struggle with the stat analysis done in this manuscript. The ANOVA output presented in the tables are totally unhelpful. There are only one set of ANOVA whereas there are 6 sets of dependent variables. So the question is what set of the dependent variables are the one ANOVA for. I must say that the readers don't need the ANOVA output but the p values. I suggest speaking with a statistician to explore the best way to do the stat analysis.

2. Still on the analysis, although as pointed out earlier, only set of p values are provided, and it is unclear what those p values are for - i hope that this is addressed. But looking at those p values provided, I notice that only the main effect of the factor "diet" is significant. When this is the case, the authors should rearrange the data in the table to show the main effects means. Namely provide the means for the two levels of phytase and the two levels of Ti (i.e. with or without Ti). Of course, if there are any interaction of factors for any of the dependent variables, the simple (all 8 means) should be listed and the means separated as appropriate with Tukey or similar.

3. On the matter of TiO2 source, is it really necessary to indicate the source. TiO2 is sold by chemical suppliers like Sigma Aldrich, Fisher etc, not the feed manufacturer.

7. PLOS authors have the option to publish the peer review history of their article (what does this mean?). If published, this will include your full peer review and any attached files.

Reviewer #1: No

---

## [Author Response · Author response to Decision Letter 1]

21 Mar 2023

20 March 2023

To the Editor

Dear Aziz ur Rahman Muhammad,

Please consider our revised manuscript for publication in PLOS ONE. Our detailed response to the comments arising from review of our previous revision (R1) are given below where specific revisions have been requested.

 Dear Dr. Brearley,

Thank you for submitting your manuscript to PLOS ONE. After careful consideration, we feel that it has merit but does not fully meet PLOS ONE’s publication criteria as it currently stands. Therefore, we invite you to submit a revised version of the manuscript that addresses the points raised during the review process.

Dear Authors, 

I still believe that manuscript require to improve the statistical part as suggested by reviewer. after improving statistical part, manuscript will be ready for publication. I invite authors to address the issues of reviewer please. 

We look forward to receiving your revised manuscript.

Kind regards,

Aziz ur Rahman Muhammad

Academic Editor

PLOS ONE

Additional Editor Comments (if provided):

Dear Authors,

I still believe that authors are required to improve the statistical part as suggested by reviewer. after improving statistical part, manuscript will be ready for publication. I invite authors to address the issues of reviewer please.

Reviewers' comments:

Reviewer's Responses to Questions

Comments to the Author

1. If the authors have adequately addressed your comments raised in a previous round of review and you feel that this manuscript is now acceptable for publication, you may indicate that here to bypass the “Comments to the Author” section, enter your conflict of interest statement in the “Confidential to Editor” section, and submit your "Accept" recommendation.

Reviewer #1: (No Response)

2. Is the manuscript technically sound, and do the data support the conclusions?

Reviewer #1: Partly

3. Has the statistical analysis been performed appropriately and rigorously? 

Reviewer #1: No

4. Have the authors made all data underlying the findings in their manuscript fully available?

Reviewer #1: Yes

5. Is the manuscript presented in an intelligible fashion and written in standard English?

Reviewer #1: No

6. Review Comments to the Author

Reviewer #1: 1. I still struggle with the stat analysis done in this manuscript. The ANOVA output presented in the tables are totally unhelpful. There are only one set of ANOVA whereas there are 6 sets of dependent variables. So the question is what set of the dependent variables are the one ANOVA for. I must say that the readers don't need the ANOVA output but the p values. I suggest speaking with a statistician to explore the best way to do the stat analysis.

2. Still on the analysis, although as pointed out earlier, only set of p values are provided, and it is unclear what those p values are for - i hope that this is addressed. But looking at those p values provided, I notice that only the main effect of the factor "diet" is significant. When this is the case, the authors should rearrange the data in the table to show the main effects means. Namely provide the means for the two levels of phytase and the two levels of Ti (i.e. with or without Ti). Of course, if there are any interaction of factors for any of the dependent variables, the simple (all 8 means) should be listed and the means separated as appropriate with Tukey or similar.

Dear Aziz ur Rahman Muhammad,

Please consider our revised manuscript for publication in PLOS One. The manuscript now bears the additional author, Philip T. Leftwich, for his contribution to the statistical analysis of data described in the manuscript. 

To clear up any further confusion, previous reference to 2 x 4 factorial trial design has been replaced with simple reference to there being 8 treatments. For ease of understanding for the reader, a table has now been included in the Materials and Methods further describing each diet/treatment (Table 1).

We have reported the variance explained by the interaction terms, and as this was a non-significant result, have dropped this from the final model in favour of an additive two-way ANOVA. In order to simplify the presentation of results, we have reported ANOVA results in text (with the necessary F values and degrees of freedom required to accurately interpret p-values) and have replaced the sum of squares table with a presentation of means and standard errors for all treatment groups. As Ti was not a significant effect, we elected to show mean summaries for all 8 treatment groups.

Additionally, tables have been simplified. They now show the means and SEM for all treatments. Means were tested by Two-Way ANOVA – as interaction was found to be not significant, means were tested again with interaction removed from the model to generate the p values presented. Follow up Tukey tests on diets were not reported in this manuscript as the effect of phytase and inositol supplementation compared to the Control for these data sets has been reported previously in Sprigg et al. (2022), and the effect of phytase was not the focus of this report. Consequently, ANOVA values have been removed from the results tables (Tables 2 & 3) and relevant information has been included in the narrative (Results section). 

3. On the matter of TiO2 source, is it really necessary to indicate the source? TiO2 is sold by chemical suppliers like Sigma Aldrich, Fisher etc, not the feed manufacturer. This author felt it appropriate to include the supplier of the TiO2 as it differentiates from titanium manufactured and sold as a fixed particle size specifically for use in purifying phosphopeptides in cell biology research (Titansphere) from mixed particle size TiO2 supplied for use as a digestibility index marker (purchased through Target Feeds with no listed manufacturer).

Yours sincerely

Charles Brearley

7. PLOS authors have the option to publish the peer review history of their article (what does this mean?). If published, this will include your full peer review and any attached files.

Do you want your identity to be public for this peer review? For information about this choice, including consent withdrawal, please see our Privacy Policy.

Reviewer #1: No

---

## [Decision Letter · Decision Letter 2]

6 Apr 2023

Accentuating the positive and eliminating the negative: efficacy of TiO2 as digestibility index marker for poultry nutrition studies

PONE-D-22-30456R2

Dear Dr. Brearley,

We’re pleased to inform you that your manuscript has been judged scientifically suitable for publication and will be formally accepted for publication once it meets all outstanding technical requirements.

Kind regards,

Aziz ur Rahman Muhammad

Academic Editor

PLOS ONE

Additional Editor Comments (optional):

Dear Authors

Thanks for revising the manuscript as suggested by reviewers.

Thanks

Reviewers' comments:

Reviewer's Responses to Questions

**Comments to the Author**

1. If the authors have adequately addressed your comments raised in a previous round of review and you feel that this manuscript is now acceptable for publication, you may indicate that here to bypass the “Comments to the Author” section, enter your conflict of interest statement in the “Confidential to Editor” section, and submit your "Accept" recommendation.

Reviewer #1: All comments have been addressed

Reviewer #3: (No Response)

2. Is the manuscript technically sound, and do the data support the conclusions?

Reviewer #1: Yes

Reviewer #3: Yes

3. Has the statistical analysis been performed appropriately and rigorously? 

Reviewer #1: I Don't Know

Reviewer #3: Yes

4. Have the authors made all data underlying the findings in their manuscript fully available?

Reviewer #1: Yes

Reviewer #3: Yes

5. Is the manuscript presented in an intelligible fashion and written in standard English?

Reviewer #1: Yes

Reviewer #3: Yes

6. Review Comments to the Author

Reviewer #1: I believe that the work done is of interest and I realize that a lot of thoughts went into the work.

Reviewer #3: (No Response)

7. PLOS authors have the option to publish the peer review history of their article (what does this mean?). If published, this will include your full peer review and any attached files.

Reviewer #1: No

Reviewer #3: No

---

## [Editor Report · Acceptance letter]

15 Jun 2023

PONE-D-22-30456R2 

Accentuating the positive and eliminating the negative: efficacy of TiO2 as digestibility index marker for poultry nutrition studies 

Dear Dr. Brearley:

I'm pleased to inform you that your manuscript has been deemed suitable for publication in PLOS ONE. Congratulations! Your manuscript is now with our production department. 

Kind regards, 

on behalf of

Dr. Aziz ur Rahman Muhammad 

Academic Editor

PLOS ONE